# Tofacitinib in Treatment-Refractory Moderate to Severe Ulcerative Colitis: Real-World Experience from a Retrospective Multicenter Observational Study

**DOI:** 10.3390/jcm9072177

**Published:** 2020-07-10

**Authors:** Peter Hoffmann, Anna-Maria Globig, Anne K. Thomann, Maximilian Grigorian, Johannes Krisam, Peter Hasselblatt, Wolfgang Reindl, Annika Gauss

**Affiliations:** 1Department of Gastroenterology and Hepatology, University Hospital Heidelberg, INF 410, 69120 Heidelberg, Germany; annika.gauss@med.uni-heidelberg.de; 2Department of Medicine II, Medical Center—University of Freiburg, Faculty of Medicine, University of Freiburg, Hugstetter Straße 55, 79106 Freiburg, Germany; anna-maria.globig@uniklinik-freiburg.de (A.-M.G.); maximilian.grigorian@uniklinik-freiburg.de (M.G.); peter.hasselblatt@uniklinik-freiburg.de (P.H.); 3Department of Medicine II, University Medical Center Mannheim, Medical Faculty Mannheim, Heidelberg University, Theodor-Kutzer-Ufer 1-3, 68167 Mannheim, Germany; Anne.Thomann@medma.uni-heidelberg.de (A.K.T.); Wolfgang.Reindl@medma.uni-heidelberg.de (W.R.); 4Department of Medical Biometry, Institute of Medical Biometry and Informatics, University Hospital Heidelberg, INF 130.3, 69120 Heidelberg, Germany; krisam@imbi.uni-heidelberg.de

**Keywords:** tofacitinib, ulcerative colitis, inflammatory bowel disease, JAK inhibitor, small molecule, real-world

## Abstract

(1) Background: Tofacitinib is approved in Europe for the treatment of adults with moderately to severely active ulcerative colitis since 2018. Real-world efficacy and safety data are currently scarce. (2) Methods: We performed a retrospective multicenter study at three German tertiary outpatient clinics for inflammatory bowel diseases and included all patients who started tofacitinib therapy between August 2018 and March 2020. The primary endpoint was a combined endpoint of steroid-free clinical remission, steroid-free clinical response, or clinical response at week 8. Secondary endpoints were biochemical response at week 8, as well as steroid-free clinical remission, steroid-free clinical response or clinical response at week 24, respectively, adverse events by week 24, and need for colectomy by the end of follow-up. (3) Results: Thirty-eight patients with moderate-to-severe ulcerative colitis were included. Eleven patients (28.9%) achieved steroid-free clinical remission at week 8. Fifty-three percent of the patients were primary non-responders at week 8. Three severe adverse events (pneumonia, hospitalization for aggravation of ulcerative colitis, emergency colectomy due to colon perforation), and 12 adverse events were documented by week 8 of therapy. By the end of follow-up, seven patients (18.4%) had undergone colectomy.

## 1. Introduction

Ulcerative colitis (UC) is an incurable, chronic inflammatory disease of the large bowel whose etiology and pathogenesis have not yet been comprehensively explained [1]. Because of this, novel treatment options should be researched. In recent years, the use of biologics in UC patients has increased whereas colectomy rates have decreased [2,3]. However, many UC patients still have to undergo colectomy during their lifetime. Medical therapeutic agents in use for the treatment of moderate-to-severe UC include corticosteroids, thiopurines, cyclosporine, tacrolimus, the biologics infliximab, adalimumab, golimumab, vedolizumab, and ustekinumab, as well as the small molecule Janus kinase (JAK) inhibitor tofacitinib since 2018. Except for steroids and thiopurines, all of the above can be used for both induction of remission and maintenance therapy in UC disease courses refractory to 5-aminosalicylate treatment. Despite the increasing spectrum of anti-inflammatory medications approved for the treatment of UC, a considerable number of UC patients remains insufficiently treated. Therefore, multiple drugs with novel mechanisms of action are being tested in different phases of clinical trials. 

The pathogenesis of UC is very complex, including genetic and environmental factors. Certainly, a wide array of cytokines is involved in the mucosal inflammatory reaction in UC. JAKs represent a family of intracellular, non-receptor tyrosine kinases transferring cytokine-mediated signals via the JAK-STAT pathway [4,5]. They play essential roles in cell growth as well as survival, development and differentiation of immune cells [6]. Tofacitinib is a small molecule, oral selective inhibitor of JAK1 and JAK3 and, to a lesser extent, of JAK2 [7].

In 2012, tofacitinib was approved by the Food and Drug Administration (FDA) for the treatment of adults with moderate to severe rheumatoid arthritis. Six years later, tofacitinib was also approved by the FDA and European Medicines Agency (EMA) for the treatment of adult patients with moderately to severely active UC. The efficacy and safety of tofacitinib in the treatment of UC patients have been extensively investigated in a phase II study [8] as well as the phase III OCTAVE clinical trial program [9]. So far, no head-to-head studies dealing with the performance of tofacitinib in comparison with other approved UC therapeutics have been published.

Given the widening spectrum of medical treatment options for UC without therapeutic guidelines in which priorities of their use are clearly defined, the decisions of physicians who treat UC patients are frequently made on an individual case-by-case basis mainly prompted by their personal experience. Real-world data and real-world evidence therefore play an increasingly important role in modern health care decisions. 

Due to the very recent approval of tofacitinib in the treatment of UC, structured and fully published real-world experience on its use is still scarce. The evaluation of novel treatment options is especially important in difficult-to-treat UC patients. The present study focusses on this subgroup of patients which represents a considerable challenge in everyday clinical practice.

## 2. Materials and Methods

### 2.1. Study Design and Data Extraction

This is an uncontrolled, retrospective multicenter observational study including outpatients with moderate-to-severe UC at three German university hospitals which serve as tertiary referral centers for the treatment of inflammatory bowel diseases (IBD). The study was approved by the local Ethics Committees (Alte Glockengießerei 11/1, 69, 115 Heidelberg, protocol number: S-274/2020; Engelberger Straße 21, 79, 106 Freiburg, protocol number: 474/14; Theodor-Kutzer-Ufer 1–3, 68, 167 Mannheim, protocol number: 2014-633N-MA). 

Inclusion criteria of the study were: age ≥18 years; diagnosis of moderately to severely active UC according to ECCO criteria [10], start of tofacitinib therapy by the beginning of March 2020, and a documented follow-up of at least 8 weeks from start of tofacitinib therapy. Patients younger than 18 years and patients with a diagnosis of Crohn’s disease or indeterminate colitis were excluded. The follow-up for all patients ended on 30 April 2020. This time point was defined as the cut-off time point for data acquisition. For efficacy analyses, patients who had to discontinue tofacitinib therapy due to adverse events prior to week 8 were considered to be non-responders.

All data were retrieved from entirely electronic medical records. Demographic and clinical parameters of all eligible patients were entered into a Microsoft Excel spreadsheet. Only pseudonymized patient data were exchanged between the participating centers.

### 2.2. Definitions

Disease extent was categorized by use of the Montreal classification for UC [11]. To assess clinical disease activity, the Partial Mayo Score (PMS) was routinely determined at every patient’s visit at the participating IBD outpatient clinics [12]. Steroid-free clinical remission was defined as a PMS of ≤2 points without concomitant use of any steroid preparation [budesonide, prednis (ol) one, or methylprednisolone]. Steroid-free clinical response was defined as a PMS reduction by ≥2 points without concomitant use of any steroid preparation. Clinical response was considered if the PMS improved by ≥2 points without an increase in steroid doses [13]. 

Mucosal healing (MH) was defined as Mayo endoscopic subscore of ≤1. Biochemical response was defined as any reduction in fecal calprotectin (FC) or plasma C-reactive protein (CRP) concentrations in comparison to baseline results.

The follow-up time was defined as the number of completed months after week 8 until 30 April, 2020, or until discontinuation of tofacitinib treatment.

### 2.3. Treatment Schedule

According to the label, tofacitinib therapy was consistently initiated at a dose of 10 mg twice per day. All patients were examined by an experienced physician at 8 weeks following the start of tofacitinib treatment. The decision of whether the dose of tofacitinib was to be reduced to 5 mg twice daily at week 8 was based on individual risk profiles of thromboembolic complications, response to therapy, and concomitant steroid medication. The decision to discontinue tofacitinib therapy due to inadequate response or adverse events was in all cases made by a senior gastroenterologist.

### 2.4. Study Endpoints

The primary study endpoint was a combined endpoint of the percentage of patients reaching steroid-free clinical remission, steroid-free clinical response or clinical response at 8 weeks of tofacitinib therapy. Secondary study endpoints were steroid-free clinical remission, steroid-free clinical response or clinical response at week 24, biochemical response at week 8, the occurrence of adverse events, and discontinuation of tofacitinib therapy due to of inadequate response or adverse events by week 24 with need for colectomy by the end of follow-up.

### 2.5. Data Collection

Further information retrieved from electronic patient charts included gender; age at data acquisition and at time of first diagnosis of UC; disease duration; disease extent; family history of IBD; presence of extraintestinal manifestations; presence of cardiopulmonary disease; cigarette smoker status; body mass index (BMI); history of anti-tumor necrosis factor alpha (TNFα), anti-integrin, or immunomodulator treatment; number of prior biological therapies; reason for tofacitinib treatment initiation; history of UC-related hospitalization(s) within 12 months prior to start of tofacitinib therapy; previous and concomitant IBD medications; suspected adverse events of tofacitinib therapy; blood and stool biochemical markers measured prior to and after start of tofacitinib therapy; and endoscopic findings. For practical reasons, not all patients were able to visit the respective IBD outpatient clinics at exactly 8 and 24 weeks of tofacitinib therapy. Therefore, all visits at 8 ± 2 and 24 ± 6 treatment weeks were considered in the analyses.

Baseline evaluations included results of stool samples and colonoscopy collected up to 6 weeks prior to start of tofacitinib therapy. FC concentrations measured at 8 ± 2 and 24 ± 6 weeks of tofacitinib therapy and colonoscopy findings from 8 to 30 weeks of tofacitinib treatment were included. Stool samples were either mailed or delivered directly to the IBD outpatient clinics by the patients.

FC concentrations of >2000 µg/g were recorded as 2000 µg/g. Plasma CRP concentrations <2 mg/L were recorded as 2 mg/L.

### 2.6. Statistical Analyses

Descriptive statistics were calculated as percentages for discrete variables and presented as medians with ranges, or as means with standard deviation, if the results were normally distributed. To identify potential predictors of response to therapy, the Mann–Whitney test was used for ordinal and continuous variables, and Chi-squared tests for categorical variables. Due to the exploratory nature of the trial, *p*-values are to be interpreted in a descriptive manner. Thus, no adjustment for multiple testing was performed. *p*-values < 0.05 were regarded as statistically significant. The statistical analyses were performed using Excel (Version 1908) and IBM SPSS Statistics 25 (Chicago, IL, USA). 

## 3. Results

### 3.1. Baseline Characteristics of the Included Patients

In total, 38 UC patients from three different treatment centers were included in the study. The median follow-up time in this study was 4 months (range: 0–18 months). The patients’ baseline characteristics are displayed in Table 1. The majority of included patients was male (68%). Median age of the patients at tofacitinib treatment initiation was 33 years, ranging from 19 to 65 years, with a median disease duration of 4 years (0–24 years). The majority of patients suffered from extensive colitis (65.7%). Thirty-two percent among them suffered from at least one extraintestinal manifestation of their UC. A concomitant cardiovascular diagnosis was documented in 10% of the patients. Rates of prior therapies with immunomodulators, anti-TNFα, and vedolizumab were 78.9%, 89.5%, and 68.4%, respectively. Only one patient (2.6%) was naïve to any biologic therapy prior to tofacitinib treatment initiation, while 26.3% of patients had received one biologic, 44.7% two biologics, 10.5% three biologics, and 15.8% four biologics. Sixteen percent of the cohort had been hospitalized due to their UC at any time during a 12-months interval prior to the start of tofacitinib treatment. 

### 3.2. Disease Activity at Baseline and Reasons for Starting Tofacitinib Therapy

The mean PMS at start of tofacitinib treatment was 6.1 (± 2.4). FC concentrations were available in 22 patients; the median FC concentration at baseline was 800 µg/g, ranging from 47 to 2000 µg/g. The median CRP plasma concentration at baseline was 8.2 mg/L, ranging from 2.0 to 115.1 mg/L (*n* = 33). Endoscopic results at baseline were available for 10 of the study participants (26.3%). The vast majority of the included patients started tofacitinib therapy due to their clinical disease activity (86.8%) (Table 1).

### 3.3. Concomitant IBD Medications at Start of Tofacitinib Therapy

Sixty-eight percent of the patients started tofacitinib while they were on 5-aminosalicylates at the same time, and 55.3% were on corticosteroids, including budesonide. Only one patient was on concomitant therapy with an immunomodulator (2.6%) (Table 1). 

### 3.4. Tofacitinib Dosing

Among the 30 patients continuing tofacitinib after week 8, 24 patients (80.0%) remained on a dose of 10 mg twice daily, while 6 patients (20.0%) continued with a dose of 5 mg twice daily.

Among the 19 patients continuing tofacitinib after week 24, 11 patients (57.9%) continued with a dose of 10 mg twice daily, whereas 8 patients (42.1%) received 5 mg twice daily.

### 3.5. Primary Study Endpoint

By week 8 of tofacitinib treatment, eleven patients (28.9%) achieved steroid-free clinical remission, five patients (13.2%) achieved steroid-free clinical response, and 2 patients (5.3%) were clinical responders, while 20 patients (52.6%) were non-responders (Figure 1a).

Four among the 38 patients (10.5%) discontinued tofacitinib therapy prior to week 8. In one patient, therapy was discontinued after 3 weeks due to missing relief of arthralgia, of which he was suffering as an extraintestinal manifestation, in one patient after 4 weeks due to spontaneous colon perforation, and in 2 patients at weeks 2 and 6 for flaring UC, respectively. At week 8, the therapy was discontinued in 4 more patients due to nonresponse. At week 8 of tofacitinib treatment, 16 patients (42.1%) were on continued corticosteroids, and 20 patients (52.6%) were on concomitant 5-aminosalicylate therapy. 

### 3.6. Secondary Study Endpoints

Among the 30 remaining patients at 8 weeks of tofacitinib therapy, 19 completed week 24, whereas one patient was lost to follow-up, and one patient did not reach 24 weeks of tofacitinib therapy by the end of data acquisition. These two patients were censored at week 8 of therapy. Figure 2 presents the Kaplan–Meier curve of time points when patients stopped tofacitinib treatment up to week 24, when 53% of the patients continued on tofacitinib therapy.

Thus, data from 36 patients were included in the statistical analyses at week 24: seven patients (19.4%) were in steroid-free clinical remission, four patients (11.1%) achieved steroid-free clinical response, two patients achieved clinical response (5.6%), and 23 patients (63.9%) were non-responders (Figure 1b).

Results of plasma CRP concentrations were available in 28 patients at week 0 and 8. In week 8, plasma CRP concentrations were either unchanged or lower compared to baseline in 82% of the patients, while FC concentrations were decreased as compared to baseline in 80% of the patients (*n* = 15).

Among the 13 patients who discontinued tofacitinib therapy prior to week 24 (all of them being non-responders), six underwent colectomy by the end of data acquisition, resulting in a total of seven colectomies by the end of April 2020.

Between week 8 and 30, eleven colonoscopies were available: five were performed in patients belonging to the group of steroid-free clinical remission, and six in patients belonging to the nonresponse group (Table 2). MH was documented in three patients of the remission group (60.0%), and two patients of the nonresponse group (33.3%).

### 3.7. Differences between Patients in Steroid-Free Clinical Remission and Those with Non-Remission at Week 8 of Tofacitinib Therapy

The biochemical parameters of interest determined at 8 ± 2 weeks of tofacitinib treatment did not differ significantly between the steroid-free remission and the non-remission group (Table 2). Due to the definition of the primary endpoint, PMS indices at week 8 were significantly lower in the remission versus the non-remission group. In contrast, the rate of patients with MH was not significantly different between the groups (Table 2). Mean BMI was significantly higher in the steroid-free clinical remission group versus the non-remission group (30.8 vs. 23.8 kg/m^2^) (Table 3). Patients who were not on concomitant steroid therapy at baseline had a greater likelihood of remaining steroid-free at week 8 of tofacitinib therapy as compared to those who received steroids at start of tofacitinib therapy.

### 3.8. Safety Profile

In Table 4, adverse events by weeks 8 and between 8 and 24 of tofacitinib therapy are listed separately. The percentages of patients experiencing at least one adverse event under tofacitinib therapy varied from 39.5% between week 0 to 8 to 52.6% between week 8 and 24. The most frequently documented adverse events were upper respiratory tract infections. From start of tofacitinib therapy until week 8, three serious adverse events were recorded: one pneumonia due to parainfluenza virus, one hospitalization due to exacerbation of UC, and one emergency colectomy due to spontaneous sigmoid perforation. In the latter patient, tofacitinib therapy was initiated as rescue therapy for severely active and refractory UC after the patient had refused surgery. None of the patients suffered from a thromboembolic complication over the study duration up to week 24.

## 4. Discussion 

The aim of the present study was to evaluate the efficacy and safety of the JAK inhibitor tofacitinib in the treatment of refractory moderately to severely active UC in the daily practice of three tertiary referral centers for IBD patients in southwest Germany. 

Our key finding is that in a treatment-refractory cohort of UC patients, steroid-free clinical remission was achieved by 28.9% of patients at 8 weeks of tofacitinib therapy, and by 19.4% of patients at week 24. About 53% of the patients were primary non-responders. 

The efficacy of tofacitinib in the treatment of UC was proven in the OCTAVE induction 1 and 2 trials which included 598 and 541 patients, respectively. The patients suffered from moderately to severely active ulcerative colitis. Fifty to sixty percent of the included patients had previously experienced anti-TNFα treatment failure, while in 60 to 70% of the patients, therapy with a classic immunosuppressant had failed [9]. 

In the OCTAVE Induction 1 trial, 18.5% of the UC patients who were treated with tofacitinib achieved clinical remission by week 8, versus 8.2% of patients who received placebo [9]. In the OCTAVE Induction 2 trial, remission at week 8 occurred in 16.6%, versus 3.6% with placebo. Very recently, a real-world observational study on tofacitinib in UC was published by the Dutch initiative on Crohn and Colitis [14]. It included 123 UC patients, of whom 95% were anti-TNFα-, 62% vedolizumab-, and 3% ustekinumab-experienced. The study endpoints were corticosteroid-free clinical, biochemical, and combined corticosteroid-free and biochemical remission at 24 weeks; the corresponding rates being 29%, 25%, and 19%, respectively. Further, in a British multicenter retrospective observational cohort study from 4 centers which included 134 UC patients, 83% of the patients had previously received at least one biologic. Overall, 74% of patients responded to tofacitinib at week 8, and steroid-free clinical remission was observed in 44% of the patients at week 26 [15]. In addition, a French cohort study on the real-world effectiveness and safety of tofacitinib in 38 patients was published: steroid-free clinical remission was observed in 34% of the patients at week 48 [16]. Colectomy-free survival was 77% at 24 weeks and 70% at 48 weeks. 

Comparable to the above-cited real-world studies, our study population was relatively treatment-refractory and therefore appears a priori more difficult to treat than the patients included in the OCTAVE trials. This is one reason why data from real-world studies are essential for treatment decisions. Our main result of achievement of steroid-free clinical remission at 8 weeks in 28.9% of the patients appears to be favorable in comparison to the results from the phase III trials. Unfortunately, the comparison of our results with those of the other European real-world studies, as well as in the OCTAVE trials in which endoscopic assessment was part of the outcome, is hampered by the fact that study endpoints were not consistent, which is a common difficulty to be faced with real-world studies. Interestingly, even though the goal of steroid-free clinical remission is more difficult to reach than clinical remission (being the primary endpoint in the phase III trials), the rates of steroid-free remission in our and the other real-world studies cited above seem to be higher than those reported for clinical remission in the phase III trials. The most obvious explanation for this may be that in observational studies, inclusion of patients is not as strictly regulated, and steroid doses were at the full discretion of the treating physicians. This implies that even at start of therapy, some of the patients had no or no relevant clinical disease activity score due to the concomitant intake of steroids. This interference of steroid therapy with the results is likely more relevant at week 8 than week 24, as according to guideline recommendations, prolonged steroid treatment phases are routinely avoided by all included treating centers in our study. Thus, the lower rate of steroid-free remission at week 24 might be more robust than the one reported for week 8. 

At week 8 only 5 of the 11 patients with steroid-free clinical remission continued with the reduced dose of 5 mg twice daily while 6 patients remained on the 10 mg twice daily dose of tofacitinib. In these cases, due to tapering off steroids during the eight weeks and patients were shortly in steroid-free clinical remission, the dose was continued with 10 mg twice daily.

One of the reasons for the primary endpoint at week 8 of tofacitinib therapy was the better comparability of the results with results from the phase III studies, and the fact that in clinical practice, treatment decisions in severely ill patients under steroid therapy cannot usually be postponed by more than 8 to 16 weeks. We found that biochemical parameters of UC disease activity did not differ significantly between weeks 0 and 8. This may be explained by the fact that about 42% of the patients were continuing on concomitant steroids at week 8. Endoscopic results revealed no higher MH rates in the steroid-free clinical remission group as compared with the non-remission group at week 8 of tofacitinib therapy. Three aspects may explain these results: (1) the above-mentioned interference of steroid therapy with effects, (2) the fact that endoscopic results were spread over a time range from week 8 until week 30 of tofacitinib therapy, and (3) the fact that overall, endoscopies were rarely performed in our cohort, although standard procedure is to assess response by colonoscopy, because in real-world settings, patients are usually not as willing to undergo endoscopic examinations as they are in prospective trials. 

With a widening spectrum of approved UC therapies producing comparable outcomes, it is of great interest to move on to more individually tailored treatment concepts. This implies that it would also be important to identify clinical or demographic factors predicting response to certain therapeutic agents. Unfortunately, due to the relatively small number of patients included in our study, regression analyses for the identification of predictors of response were not statistically feasible. However, in our study, we found hints that a low BMI may be a suitable parameter to predict nonresponse to tofacitinib therapy in treatment-refractory UC patients but could also represent a statistical anomaly. 

In the therapy of treatment-refractory UC patients, a stringent endpoint is colectomy-free survival. In our study, 18.4% of the patients had undergone colectomy by the end of data acquisition. This rate of nearly a fifth is fairly close to colectomy rates reported for inpatients with steroid-refractory severe acute ulcerative colitis, reaching 70 to 80% after 6 to 12 months [17]. In addition to the fact that 97% of the patient cohort of this study had experienced at least one biologic therapy and 71% had received at least 2 biologics prior to the start of tofacitinib therapy, the relatively high colectomy rate demonstrates that the patients included in our study belong to a treatment-resistant group, so that the results do not reflect the treatment outcomes to be expected for the UC population as a whole. 

The main documented adverse events were upper respiratory tract infections with 15.8% and 26.3% to week 8 and 24. This rate is higher than in the OCTAVE trials where nasopharyngitis occurred in up to 13.8%. Nevertheless, it was one of the most frequent adverse events in our study as well as in the OCTAVE trials. 

No thromboembolic events were observed in our study, even though 80.6 percent of the patients received the higher tofacitinib dose of 20 mg per day after the initiation phase. Of course, the follow-up of this study is too short for a reliable statement on tofacitinib treatment as a potential thrombotic risk factor in UC. However, the question whether tofacitinib may trigger thromboembolic events in UC patients as in patients with rheumatoid arthritis remains very important, because more than half of the patients who were still on tofacitinib at week 24 in our study were treated with the high dose of 20 mg tofacitinib per day. 

The main strength of our study lies in the homogeneity of the observed study population, representing a treatment-refractory patient group as typically found at IBD outpatient clinics of German university hospitals, where difficult-to-treat IBD patients are referred to. The participating study centers are part of a network whose physicians and study nurses meet in regular intervals to harmonize data acquisition. All centers operate IBD registries for the prospective inclusion of patients receiving new therapies; the clinical variables of interest and the time points of data acquisition were uniform. The primary study endpoint of steroid-free clinical remission at 8 weeks in this study is ambitious, yet it reflects the goal that is ideally set in the interest of the patients. 

A major limitation of the study is the relatively small number of included patients, which is partly explained by the tofacitinib safety alert issued in March 2019 by the FDA, warning that treatment with tofacitinib at a dose of 10 mg twice daily is associated with an increased risk for pulmonary embolism and death in patients with rheumatoid arthritis. At the treating centers, this warning resulted in an abrupt reduction in tofacitinib initiation rates; after March 2019, only seven more refractory patients were started on the JAK inhibitor. Another limitation of the study is that endoscopic data were only available in a small proportion of patients. 

In conclusion, eleven patients (28.9%) achieved steroid-free clinical remission at week 8 of tofacitinib therapy in a treatment-refractory real-world cohort of patients suffering from moderately to severely active UC. The safety profile was acceptable in the observed time frame.

## Figures and Tables

**Figure 1 jcm-09-02177-f001:**
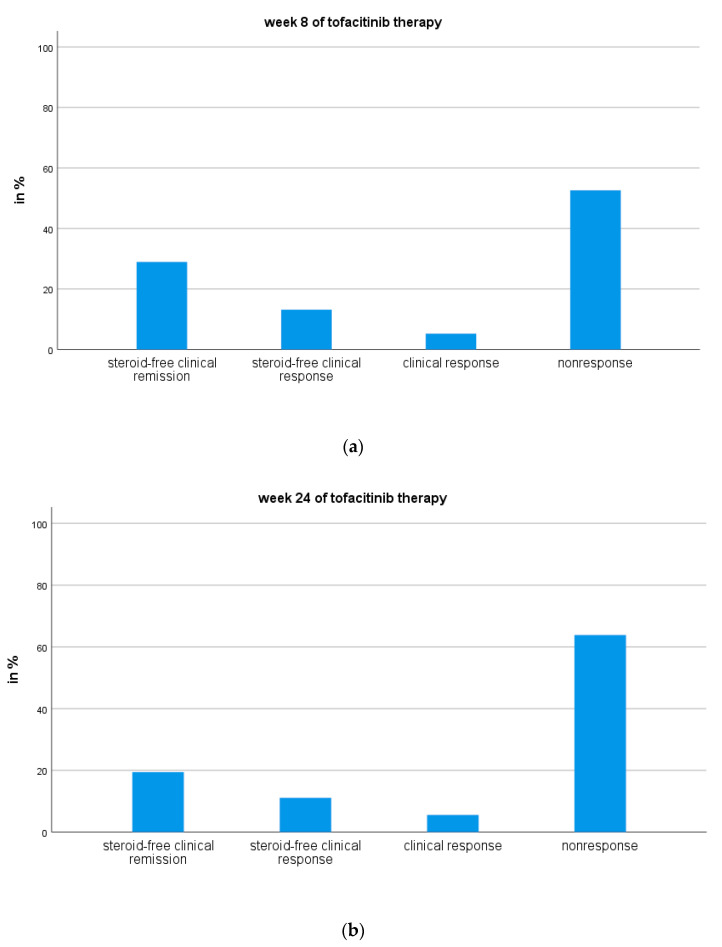
Tofacitinib therapy outcomes at weeks 8 (*n* = 38) and 24 (*n* = 36). (**a**) week 8 of tofacitinib therapy; (**b**) week 24 of tofacitinib therapy.

**Figure 2 jcm-09-02177-f002:**
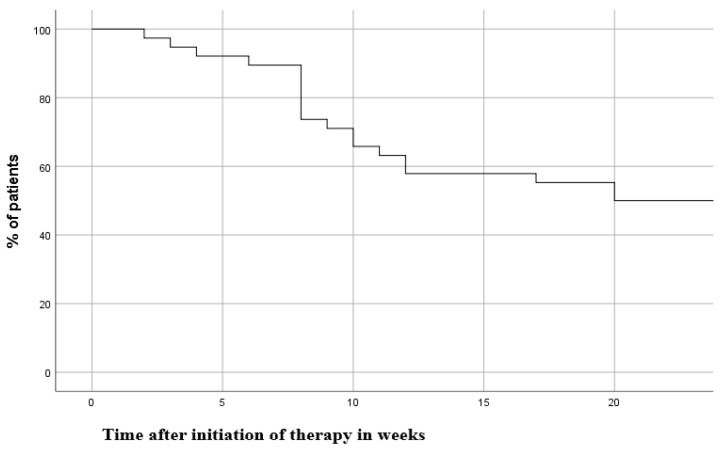
Kaplan–Meier curve of discontinuation of tofacitinib therapy.

**Table 1 jcm-09-02177-t001:** Baseline characteristics of the included patients.

Variable	*n* = 38
Male, *n* (%)	26 (68.4)
Age at diagnosis (years), median (range)	27 (12–63)
Age at start of treatment (years), median (range)	33 (19–65)
**Montreal Classification of UC:**	
Age, *n* (A1:A2:A3)	7:23:8
Location, *n* = 35 (E1:E2:E3)	1:11:23
Prior intestinal resections or appendectomy, (%)	0 (0)
First degree relative(s) with IBD, *n* = 36 (%)	4 (11.1)
Disease duration at baseline (years), median (range)	4 (0–24)
Diagnosis of at least one extraintestinal manifestation, *n* (%)	12 (31.6)
Diagnosis of cardiopulmonary disease, *n* (%)	4 (10.5)
History of colorectal carcinoma, *n* (%)	0 (0)
Active cigarette smoking, *n* = 37 (%)	2 (5.4)
BMI (kg/m^2^), mean ± SD (range), *n* = 37	24.5 ± 5.1 (16.6–40.0)
History of total hospitalizations within 12 months to baseline, *n* (%)	8 (21.1)
History of UC-related hospitalizations within 12 months to baseline, *n* (%)	6 (15.8)
History of anti-TNFα treatment, *n* (%)	34 (89.5)
History of anti-integrin treatment, *n* (%)	26 (68.4)
History of immunomodulator treatment, *n* (%)	30 (78.9)
**Prior Exposure to Biologic(s):**	
0 biologics, *n* (%)	1 (2.6)
1 biologic, *n* (%)	10 (26.3)
2 biologics, *n* (%)	17 (44.7)
3 biologics, *n* (%)	4 (10.5)
4 biologics, *n* (%)	6 (15.8)
**Reason for Initiating Tofacitinib Treatment**	
Clinical disease activity, *n* (%)	33 (86.8)
Endoscopy results, *n* (%)	1 (2.6)
High FC concentration, *n* (%)	3 (7.9)
Intolerance to prior therapy, *n* (%)	1 (2.6)
**Concomitant IBD Medications at Baseline:**	
Mesalazine/sulfasalazine, *n* (%)	26 (68.4)
Steroids (including budesonide), *n* (%)	21 (55.3)
Immunomodulators, *n* (%)	1 (2.6)
PMS, mean ± SD (range)	6.1 ± 2.4 (0–9)
**Endoscopic Disease Activity at 0–6 Weeks to Baseline (*n* = 10):**	
Mayo Score I, *n* (%)	1 (10)
Mayo Score II, *n* (%)	4 (40)
Mayo Score III, *n* (%)	5 (50)
**Biochemical Parameters at Baseline:**	
Plasma CRP concentration (mg/L), median (range), *n* = 33	8.2 (2.0–115.1)
WBC count, (/nL), median (range), *n* = 35	10.0 (5.6–17.8)
Hemoglobin concentration (g/dL), mean ± SD (range), *n* = 36	12.9 ± 2.4 (6.4–17.2)
PLT count (/nL), mean ± SD (range), *n* = 36	390 ± 154 (147–730)
Plasma albumin concentration (g/L), mean ± SD (range), *n* = 28	42.3 ± 5.4 (26.0–49.3)
FC concentration (µg/g), median (range), *n* = 22	800 (47–2000)

BMI: body mass index; CRP: C-reactive protein; FC: fecal calprotectin; IBD: inflammatory bowel disease; PLT: platelet; PMS: Partial Mayo Score; SD: standard deviation; TNFα: tumor necrosis factor alpha; UC: ulcerative colitis; WBC: white blood cell.

**Table 2 jcm-09-02177-t002:** Comparison between parameters determined at 8 ± 2 weeks from start of tofacitinib therapy between the group reaching steroid-free clinical remission and the non-remission group.

	Steroid-Free Clinical Remission	Non-Remission	*p*-Value
*n* = 38 (%)	11 (28.9)	27 (71.1)	
PMS, mean ± SD (range), *n* = 34	1.0 ± 1.0 (0–2)	4.8 ± 2.0 (0–9) (*n* = 23)	<0.01 ^2^
BMI (kg/m^2^), mean ± SD (range), *n* = 30	30.8 ± 6.7 (19.6–41.1) (*n* = 7)	23.8 ± 3.3 (18.0–29.8) (*n* = 23)	<0.01 ^2^
**Endoscopic Findings Between 8 and 30 Weeks**			
Colonoscopy, *n* = 11	5	6	
Mucosal healing, *n* (%)	1 (20.0)	1 (16.7)	0.89 ^1^
**Biochemical Parameters:**			
Plasma CRP concentration (mg/L), median (range), *n* = 33	3.0 (2.0–26.0)	5.3 (2.0–32.1) (*n* = 22)	0.67 ^2^
WBC count (/nL), median (range), *n* = 33	8.6 (4.8–11.6)	8.3 (5.7–13.3) (*n* = 22)	0.75 ^2^
Hemoglobin concentration (g/dL), mean ± SD (range), *n* = 33	12.9 ± 1.9 (9.5–15.8)	13.7 ± 1.9 (8.6–17.7) (*n* = 22)	0.27 ^2^
PLT count (/nL), mean ± SD (range), *n* = 33	318 ± 100 (151–461)	366 ± 154 (170–768) (*n* = 22)	0.49 ^2^
Plasma albumin concentration (g/L), mean ± SD (range), *n* = 28	43.9 ± 5.4 (32.5–50.4) (*n* = 9)	45.1 ± 2.3 (41.3–48.8) (*n* = 19)	0.79 ^2^
FC concentration (µg/g), median (range), *n* = 21	368 (39–1800) (*n* = 6)	300 (76–1800) (*n* = 15)	0.73 ^2^
Reduced FC concentration and CRP at week 8 compared to baseline, *n* = 13 (%)	2 (50)	6 (66.7)	0.57 ^1^

BMI: body mass index; CRP: C-reactive protein; FC: fecal calprotectin; PLT: platelet; PMS: Partial Mayo Score; WBC: white blood cell; ^1^ Chi-squared test; ^2^ Mann–Whitney-test.

**Table 3 jcm-09-02177-t003:** Comparison of baseline characteristics between the subgroups of patients with steroid-free clinical remission versus non-remission at week 8 of tofacitinib therapy.

Parameter	Steroid-Free Clinical Remission	Non-Remission	*p*-Value
*n* = 38 (%)	11 (28.9)	27 (71.1)	
Male, *n* (%)	6 (54.5)	20 (74.1)	0.24 ^1^
Age at diagnosis (years), median (range)	27.0 (15–51)	25.0 (12–63)	0.86 ^2^
Age at baseline (years), median (range)	36.0 (20–57)	31.0 (19–65)	0.17 ^2^
**Montreal Classification of UC:**			
Age, *n* (A1:A2:A3); *n* = 35	2:6:3	5:2:7	0.83 ^1^
Location, *n* (E1:E2:E3)	1:3:7	0:8:16	0.32 ^1^
First degree relative(s) with IBD, *n* (%), *n* = 36	1 (11.1)	3 (11.1)	1.00 ^1^
Disease duration at baseline (years), median (range)	5 (2–20)	4 (0–24)	0.08 ^2^
Presence of at least one extraintestinal manifestation, *n* (%)	3 (27.3)	9 (33.3)	0.72 ^1^
Presence of cardiopulmonary disease, *n* (%)	1 (9.1)	3 (11.1)	0.86 ^1^
Active cigarette smoking, *n* = 37 (%)	1 (10.0)	1 (3.7)	0.45 ^1^
BMI (kg/m^2^), median (range), *n* = 37	28.1 (18.6–40.0)	23.1 (16.6–29.8) (*n* = 26)	0.03 ^2^
History of total hospitalizations within 12 months to baseline, *n* (%)	2 (18.2)	6 (22.2)	0.78 ^1^
History of UC-related hospitalizations within 12 months to baseline, *n* (%)	2 (18.2)	4 (14.8)	0.80 ^1^
History of anti-TNFα treatment, *n* (%)	11 (100.0)	23 (85.2)	0.18 ^1^
History of anti-integrin treatment, *n* (%)	8 (72.7)	18 (66.7)	0.72 ^1^
History of immunomodulator treatment, *n* (%)	10 (90.9)	20 (74.1)	0.25 ^1^
PMS at baseline, median (range)	7 (2–8)	7 (0–9)	0.48 ^2^
**Prior Exposure to Biologics:**			0.15 ^1^
0 biologics, *n* (%)	0 (0)	1 (3.7)	
1 biologic, *n* (%)	1 (9.1)	9 (33.3)	
2 biologics, *n* (%)	4 (36.4)	13 (48.1)	
3 biologics, *n* (%)	2 (18.2)	2 (7.4)	
4 biologics, *n* (%)	4 (36.4)	2 (7.4)	
**Concomitant Medication at Baseline:**			
Mesalazine/sulfasalazine, *n* (%)	6 (54.5)	20 (74.1)	0.24 ^1^
Steroids (including budesonide), *n* (%)	3 (27.3)	18 (66.7)	0.03 ^1^
Immunomodulators, *n* (%)	1 (9.1)	0 (0)	0.11 ^1^
**Biochemical Parameters at Baseline:**			
Plasma CRP concentration (mg/L), median (range), *n* = 33	7.8 (2.0–62.0) (*n* = 10)	8.2 (2.0–115.1) (*n* = 23)	0.78 ^2^
WBC count (/nL), median (range), *n* = 35	8.8 (6.3–17.5) (*n* = 10)	10.4 (5.6–17.8) (*n* = 25)	0.54 ^2^
Hemoglobin concentration (g/dL), median (range), *n* = 36	13.4 (6.4–15.3)	13.5 (7.9–17.2) (*n* = 25)	0.72 ^2^
PLT count (/nL), median (range), *n* = 36	339 (180–730)	346 (147–707) (*n* = 25)	0.44 ^2^
Plasma albumin concentration (g/L), median (range), *n* = 28	42.0 (36.8–47.9) (*n* = 7)	43.5 (26.0–49.3) (*n* = 21)	0.85 ^2^
FC concentration (µg/g), median (range), *n* = 22	800 (384–2000) (*n* = 6)	816 (47–1800) (*n* = 16)	0.68 ^2^

BMI: body mass index; CRP: C-reactive protein; FC: fecal calprotectin; PLT: platelet; PMS: Partial Mayo Score; SD: standard deviation; TNFα: tumor necrosis factor alpha; UC: ulcerative colitis; WBC: white blood cell. ^1^ Chi-squared test; ^2^ Mann–Whitney-test.

**Table 4 jcm-09-02177-t004:** Adverse events in the study cohort listed according to the time of their occurrence.

Therapy Weeks	0–8	8–24
*n*	38	19
Serious adverse events, *n* (%)	3 (7.9)	0
Viral pneumonia, *n* (%)	1 (2.6)	
Worsening of UC, *n* (%)	1 (2.6)	
Colon perforation, *n* (%)	1 (2.6)	
Adverse events, *n* (%)	12 (31.6)	10 (52.6)
Fungal skin infection, *n* (%)	1 (2.6)	
Dizziness, *n* (%)	1 (2.6)	
Arthralgia, *n* (%)		1 (5.3)
Headaches, *n* (%)	1 (2.6)	1 (5.3)
Upper respiratory tract infection, *n* (%)	6 (15.8)	5 (26.3)
Fever of unknown origin, *n* (%)		2 (10.5)
Influenza, *n* (%)	1 (2.6)	
Flatulence, *n* (%)	1 (2.6)	
Elevated liver enzymes, *n* (%)	1 (2.6)	
Microhematuria, *n* (%)		1 (5.3)

UC: ulcerative colitis.

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
