# Peer review of "Tofacitinib in Treatment-Refractory Moderate to Severe Ulcerative Colitis: Real-World Experience from a Retrospective Multicenter Observational Study"

_jcm, 2020, doi:10.3390/jcm9072177_

Round 1

Reviewer 1 Report

Overall this is a well written review of the experience of 3 German tertiary care centres and this adds to real world experience of using Tofacitinib for moderate to severe UC. Although it is not original it does add usefully to the literature and is worthy of publication. The cohort of patients are typical of those being considered for treatment, with high proportion having one or more biologic agents used.

I have only a couple of specific comments on the paper:

  1. At week 24 64% of patients were non-responders however 61% remained on the drug, this is not unusual but shows that clinicians are sometimes more comfortable starting than stopping drugs, especially when limited options remain - this might be commented on.
  2. There are several Tables and Figures and I wonder whether Table 3 could be included as Supplementary data rather than included in the main text.
  3. In 3.8 there is a typo with sigma being sigmoid.
  4. The finding of low BMI being a predictor of non-response has been highlighted - has this been reported in any other studies to know whether this is a true effect or a Type 2 error.
  5. The concluding sentence should be altered as this study does not show efficacy as it isn't a randomised placebo-controlled study - this should be changed and just report the numbers, as in the abstract.

Author Response

1. At week 24 64% of patients were non-responders however 61% remained on the drug, this is not unusual but shows that clinicians are sometimes more comfortable starting than stopping drugs, especially when limited options remain - this might be commented on.

  • Figure Ib compares week 0 to week 24, there are not only the patients included who continued at week 8, therefore the nonresponse rate is still so high. To make this clear, the numbers are shown in the legend in the revised version of the manuscript.

2. There are several Tables and Figures and I wonder whether Table 3 could be included as Supplementary data rather than included in the main text.

  • Table 3 shows BMI as a suitable parameter to predict nonresponse to tofacitinib therapy. Therefore, we would not like to exclude Table 3 from the main text.

3. In 3.8 there is a typo with sigma being sigmoid.

  • Sigma was changed to sigmoid.

4. The finding of low BMI being a predictor of non-response has been highlighted - has this been reported in any other studies to know whether this is a true effect or a Type 2 error.

  • This could be a Type 2 error, „but could also represent a statistical anomaly.“ Was included in the discussion.#

5. The concluding sentence should be altered as this study does not show efficacy as it isn't a randomised placebo-controlled study - this should be changed and just report the numbers, as in the abstract.

  • The conclusion has been changed as suggested (page 12).

Reviewer 2 Report

This is a multicentre study of the effectiveness and safety of tofacitinib. Although there have been several cohorts that have published in this area, reports of real-world outcomes are limited and this study adds to that literature.

The study is limited by its retrospective nature and the small numbers as well as the lack of endoscopic data in many patients.

The definition of mucosal healing is unusual. Absence of ulceration is often used in Crohn’s disease, but in UC, ulceration is equivalent to a Mayo endoscopic score of 3 suggesting that in this study all other findings ( ie including Mayo 2) were regarded as being equivalent to mucosal healing. The authors should justify this. In addition, could the authors clarify why colonoscopy was performed as there is a clear potential for bias depending on whether a colonoscopy was performed routinely to assess response (regardless of clinical state) or whether it was performed because of ongoing symptoms (or some other reason)

The manuscript states that 31 patients continued tofacitinib after week 8, 25 of whom stayed on 10mg bd. However, 11 patients were in steroid-free remission by week 8. Therefore, at least 5 patients were continued on 10mg bd at week 8 despite having achieved steroid-free remission. Why was this? In addition, in section 3.4, 31 patients are described as continuing therapy after week 8, but this becomes 30 patients in the next section.

Paired data on CRP and FC are available and should be presented – was there a significant drop in FC / CRP overall or in responders vs non-responders?

The authors identify reasons why remission is relatively less common in the OCTAVE studies - a major reason they do not mention is that endoscopic assessment was part of the outcome in OCTAVE at week 8 rather than relying on PMS

The authors’ comments in the discussion about low BMI being a predictor of non-response are too strong; no corrections were made for multiple analyses and this has not come out in any other study of which I am aware. A more reasonable interpretation would be that their finding is not a hint that low BMI is a predictor of non-response, but rather that, in this small number of patients, it is most likely to represent a statistical anomaly.

For figure 1, tofacitinib therapy outcomes at weeks 8 and 24, suggest including the absolute number ‘n’ of patients for ease of interpretation.

Under section 3.5, please rephrase ‘missing relief of arthralgia’ to make it clearer why this patient discontinued tofacitinib.

Author Response

The definition of mucosal healing is unusual. Absence of ulceration is often used in Crohn’s disease, but in UC, ulceration is equivalent to a Mayo endoscopic score of 3 suggesting that in this study all other findings ( ie including Mayo 2) were regarded as being equivalent to mucosal healing. The authors should justify this. In addition, could the authors clarify why colonoscopy was performed as there is a clear potential for bias depending on whether a colonoscopy was performed routinely to assess response (regardless of clinical state) or whether it was performed because of ongoing symptoms (or some other reason)

  • The definition of mucosal healing was changed at page 3.
  • In table 2, the numbers of mucosal healing have been changed according to the new definition, and results have been changed in section 3.7, page 8.
  • On page 11 in the discussion, the interpretation of the endoscopic findings has been changed.
  • Colonoscopy is normally performed routinely to assess response, but if patients do not experience symptoms, they are not willing to undergo colonoscopy. This has been changed in the discussion page 13.

  The manuscript states that 31 patients continued tofacitinib after week 8, 25 of whom stayed on 10mg bd. However, 11 patients were in steroid-free remission by week 8. Therefore, at least 5 patients were continued on 10mg bd at week 8 despite having achieved steroid-free remission. Why was this? In addition, in section 3.4, 31 patients are described as continuing therapy after week 8, but this becomes 30 patients in the next section.

  • Indeed there was one patient who stopped tofacitinib therapy directly after the appointment at week 8 due to symptoms. So this patient was interpreted wrongly: we changed the numbers in section 3.4 accordingly.

Paired data on CRP and FC are available and should be presented – was there a significant drop in FC / CRP overall or in responders vs non-responders?

  • This data is now included in table 2. There is a considerable bias in this data because fecal calprotectin is mainly turned in by the patients if they have symptoms.

The authors identify reasons why remission is relatively less common in the OCTAVE studies - a major reason they do not mention is that endoscopic assessment was part of the outcome in OCTAVE at week 8 rather than relying on PMS

  • This point was added in the discussion page 11.

The authors’ comments in the discussion about low BMI being a predictor of non-response are too strong; no corrections were made for multiple analyses and this has not come out in any other study of which I am aware. A more reasonable interpretation would be that their finding is not a hint that low BMI is a predictor of non-response, but rather that, in this small number of patients, it is most likely to represent a statistical anomaly.

  • Due to the relatively small number of patients included in our study, regression analyses for the identification of predictors of response were not statistically feasible; This is explained on page 11.
  • The possibility of a statistical anomaly was added on page 11.

For figure 1, tofacitinib therapy outcomes at weeks 8 and 24, suggest including the absolute number ‘n’ of patients for ease of interpretation.

  • This was changed in the legend of figure 1.

Under section 3.5, please rephrase ‘missing relief of arthralgia’ to make it clearer why this patient discontinued tofacitinib.

  • This was changed to: "missing relief of arthralgia, of which he was suffering as an extraintestinal manifestation

Reviewer 3 Report

Ulcerative colitis (UC), also clinically known as chronic nonspecific UC, is a chronic intestinal inflammatory disease whose etiology and pathogenesis have not yet been comprehensively explained. To evaluate the efficacy and safety of the JAK inhibitor tofacitinib in the treatment of refractory moderately to severely active UC, Dr. Peter Hoffmann and coworkers performed a retrospective multicenter study at three German tertiary outpatient clinics for inflammatory bowel diseases. The results reveal tofacitinib proved to be efficacious in a treatment-refractory real-world cohort of patients suffering from moderately to severely active UC. The safety profile was acceptable in the observed time frame.

The style and overall representation of the manuscript are well organized, and the study pattern and details are described with high quality. However, the authors still need to make a minor correction.

Particular comments:

1) Page 1, in the author list, an asterisk should be used to give credit to corresponding author.

2) Page 2, item number “.22.2” should be corrected to “2.2”.

3) The table number was in a total mess, Table 1 was jumped to Table 4 directly.

4) Uniform the Figure number by “Figure 1” other than “Figure I”.

5) Page 13, Ref. 10, the journal name should be italic.

Author Response

Particular comments:

  • Page 1, in the author list, an asterisk should be used to give credit to corresponding author.
    • This was chanced on page 1
  • Page 2, item number “.22.2” should be corrected to “2.2”.
    • It is corrected now to „2.2“.
  • The table number was in a total mess, Table 1 was jumped to Table 4 directly.
    • The order of the tables were chanced to the correct position (Table 4 was changed to table 2 (page 7))
  • Uniform the Figure number by “Figure 1” other than “Figure I”.
    • Numbers of figures were changed to 1 and 2.
  • Page 13, Ref. 10, the journal name should be italic.
    • The journal name is now italic

Round 2

Reviewer 2 Report

I am happy with the changes - thank you for incorporating them. Just one minor comment was not addressed I think

The manuscript states that 31 patients continued tofacitinib after week 8, 25 of whom stayed on 10mg bd. However, 11 patients were in steroid-free remission by week 8. Therefore, at least 5 patients were continued on 10mg bd at week 8 despite having achieved steroid-free remission. Why was this?

I think this just needs a comment in the discussion - apologies if I have missed this

Author Response

The manuscript states that 31 patients continued tofacitinib after week 8, 25 of whom stayed on 10mg bd. However, 11 patients were in steroid-free remission by week 8. Therefore, at least 5 patients were continued on 10mg bd at week 8 despite having achieved steroid-free remission. Why was this?

It is now written in the discussion:

At week 8 only 5 of the 11 patients with steroid-free clinical remission continued with the reduced dose of 5mg twice daily while 6 patients remained on the 10mg twice daily dose of tofacitinib. In these cases, due to tapering off steroids during the eight weeks and patients were shortly in steroid-free clinical remission, the dose was continued with 10mg twice daily.